

# Digital PBL-CBL teaching method improves students' performance in learning complex implant cases in atrophic anterior maxilla

Dan Chen[1,*], Wenyan Zhao[1,*], Li Ren[1], Kunli Tao[1], Miaomiao Li[2], Beiju Su[2], Yunfei Liu[1], Chengzhe Ban[3] and Qingqing Wu[1]

[1] Stomatological Hospital of Chongqing Medical University, Chongqing Key Laboratory of Oral Diseases and Biomedical Sciences, Chongqing Municipal Key Laboratory of Oral Biomedical Engineering of Higher Education, Chongqing, China
[2] Dazhu Traditional Chinese Medicine Hospital, Chongqing, China
[3] Ruitai Stomatological Hospital, Chongqing, China
[*] These authors contributed equally to this work.

## ABSTRACT

**Background.** The clinical teaching of esthetic implant-supported restoration of the atrophic maxilla is challenging due to the complexity and unpredictability of bone and soft tissue augmentation. The traditional problem-based learning and case-based learning method (PBL-CBL method) with a full digital workflow (digital PBL-CBL method) provides the students a chance to view clinical cases in a more accurate and measurable way. The aim is to evaluate the effectiveness of the new digital PBL-CBL method in teaching complex implant cases in esthetic area.

**Materials & Methods.** A full digital workflow of dental implant therapy was established for patients with severely atrophic anterior maxilla. The digital data of cases done in the new workflow was used as teaching materials in digital PBL-CBL teaching. Fifty-four postgraduate students were randomly selected and divided into three groups, including traditional PBL-CBL group (students taught in a PBL-CBL method with no digital cases), digital PBL-CBL group (students taught in a PBL-CBL method with full digital cases) and control group (students taught in didactic teacher-centered method). After training for three months, a study of the students' opinions on the corresponding teaching method was carried out through a feedback questionnaire. A theory test was used to evaluate students' mastery of knowledge about tissue augmentation and esthetic implant restoration. A case analysis was used to determine whether students could apply the knowledge to problem solving.

**Results.** The digital PBL-CBL method resulted in a higher rate of satisfaction than the traditional PBL-CBL method and the didactic teacher-centered method in all items except for "This approach decreases extracurricular work". Case analysis scores of the digital PBL-CBL group were significantly higher than that of the traditional PBL-CBL group and the control group. For the theory test, the digital PBL-CBL group ($61.00 \pm 6.80$) but not traditional PBL-CBL group ($55.22 \pm 9.86$) obtained a significant higher score than the control group ($45.11 \pm 12.76$), although no significant difference was found between the digital PBL-CBL group and the traditional PBL-CBL group.

Corresponding authors
Chengzhe Ban,
banchengzhe@163.com
Qingqing Wu,
501190@hospital.cqmu.edu.cn

**Conclusion**. Compared with other methods, students taught with the digital PBL-CBL method showed higher satisfaction and better performance in acquisition of academic knowledge and ability in solving practical clinical problems. The digital PBL-CBL method provided a promising alternative for teaching complex implant cases at the anterior maxilla.

## INTRODUCTION

Implant-supported restoration is considered to be highly complex and risky for patients with extensive missing teeth and a severely atrophic anterior maxilla (*Rocchietta, Fontana & Simion, 2008*). To better provide functional and esthetic results, the treatment protocol for those patients often involves multiple steps, including augmenting the bone vertically and horizontally, prosthetically-oriented placement of dental implants, soft tissue management, provisional restoration, and the final restoration (*Aghaloo et al., 2016*; *Checchi et al., 2019*). During the above operations, the dentist must determine when and which step to take based on the patient's condition. Besides, each step has a considerable number of surgical and restorative techniques with different indications. The dentist must select the appropriate technique for each step, which places a high demand on professional knowledge and hands-on practice skills. Even a minor mistake may lead to compromised esthetic complications (*Ramanauskaite & Sader, 2022*). The above characteristics of implant therapy at the atrophic maxilla make it difficult for teachers to impart knowledge to students in dental education. The World Health Organization (1994) recommended that dental education should be problem-oriented to develop the comprehensive competencies of dental students (*Gerzina et al., 2003*). However, studies by *Arias et al. (2016)* and *Malu (2010)*, have shown that teaching in most medical and dental schools is predominantly teacher-centered and students tend to learn by rote, which makes teaching ineffective and cannot meet the requirement of the World Health Organization. It is necessary for dental schools to find a suitable teaching method of dental implant therapy at severely atrophic maxilla.

Learning dental implant therapy at severely atrophic maxilla requires highly creative thinking and the good imagination of the students (*Botelho, Gao & Bhuyan, 2018*; *Dut et al., 2011*). All the surgical interventions before final restoration (soft tissue manipulation, bone augmentation and implant placement) should be guided by the three-dimensional position of the restoration presenting in the final step, indicating that students must keep the visual final restoration in mind from the very start, although it will be fabricated months later (*Chackartchi et al., 2022*). Besides, the first step, the bone augmentation surgery, is not completely predictable, which is not always able to guarantee the expected result due to graft resorption, especially in the atrophic anterior maxilla (*Checchi et al., 2019*; *Hameed et al., 2019*; *Mertens et al., 2013*; *Steller et al., 2022*; *Uehara et al., 2015*). This initial step lays the foundation for the following procedures because correction of osseous

deficiencies allows ideal implant placement and creates a more natural soft tissue profile for better crown anatomy and esthetics (*Aghaloo et al., 2016*). The unpredictable resorption of first-stage bone augmentation can be compensated by an additional corrective bone augmentation or soft tissue intervention during implant placement and recovery surgery (*Checchi et al., 2019*). In fact, staged horizontal ridge augmentation with autogenous bone blocks or guided bone regeneration (GBR) entailed an additional bone augmentation procedure before implant therapy in 37% of cases (*Jensen, Jensen & Worsaae, 2016*). Given the characteristics mentioned above, it is high time to establish a teaching method that facilitates practical thinking, careful observation, and flexible decision making of dental students to prepare them for clinical work.

Problem-based learning (PBL) and case-based learning (CBL) have been described as effective and acceptable tools for medical and dental education (*Koh et al., 2008*; *Thistlethwaite et al., 2012*; *Tomaz et al., 2015*). PBL teaching method is teacher-led, student-led and patient's problem-oriented to cultivate students' ability to discover and solve problems independently, whereas the CBL teaching method is based on authentic clinical cases, which enables learners to solve similar problems point-to-point (*Thistlethwaite et al., 2012*; *Wosinski et al., 2018*). Previous studies combined PBL and CBL in teaching and found that the PBL-CBL method was more effective than PBL or CBL (*Dong & Zeng, 2017*; *Ginzburg et al., 2018*; *Liu et al., 2020*; *Yang et al., 2023*). Whether combined or applied independently, the problems and cases used in traditional PBL and CBL are usually described in abstract words or 2-dimensional pictures, while the clinical decision to make timely correction depends on accurate calculation of the 3-dimentional discrepancies of bone and soft tissue contour, which is far beyond the scope of words or pictures. It can be seen that neither the didactic teacher-centered method nor the PBL–CBL method is suitable for teaching of dental implant therapy at severely atrophic maxilla. The full digital workflow of dental implant therapy approximates the interface of prosthetic and surgical implant treatment, from the virtual planning (plotted on a guidance template) to the computer-assisted design/computer-assisted manufacturing-based design (including production of the final prosthodontic rehabilitation) (*Ben Yehuda et al., 2020*; *Chochlidakis et al., 2020*; *Papaspyridakos et al., 2020a*; *Papaspyridakos et al., 2020b*; *Rojas Vizcaya, 2018*). These digital techniques feature virtual 3D reconstruction of bone, superimposing myriad scans and precisely aligning them with common data points (*Li et al., 2021*; *Roberts, Shull & Schiner, 2020*; *Vandeweghe et al., 2017*; *Ye et al., 2020*), which offer the dental team opportunity to design the final prosthetic virtually first and make a plan of bone augmentation and implant placement precisely guided by this virtual preview. The prerequisite is the superimposition of the bone contour and intraoral scans (*Joda & Buser, 2013*), which could precisely reveal the location and volume in graft resorption, providing the students with accurate information to tell when and where additional hard or soft tissue are required, and which surgical technique is indicated. Therefore, problems or cases explained with a full digital workflow may provide the students a chance to view clinical cases in a more accurate, measurable and intuitive way.

This study details a PBL-CBL method with a full digital workflow (digital PBL-CBL method), which features vivid 3D presentation of final restoration and tissue contour,

measurable tissue contour changes, and accurate correction of contour discrepancies using digital tools. The aim of this study is to evaluate the effectiveness of the digital PBL-CBL method in teaching complex implant cases at atrophic anterior maxilla.

## MATERIAL AND METHODS

### Students and grouping

This study was conducted in accordance with the guidelines set out in the Declaration of Helsinki and approved by the Ethics Committee of the Affiliated Stomatological Hospital of Chongqing Medical University (No. KQJ2022166). Written informed consent was obtained from all participants. Fifty-four postgraduate students were included in this study, who received training between 2020 and 2023 at the Department of Implant Dentistry, the Affiliated Stomatological Hospital of Chongqing Medical University. No students had any experience or training in bone and soft tissue augmentation at atrophic maxilla before hand. The students were randomly allocated into three groups, namely the traditional PBL-CBL group (students were trained using traditional PBL-CBL method), the digital PBL-CBL group (students were trained using digital PBL-CBL method), and control group (students were trained using didactic teacher-centered method). All groups were trained for three months.

### Establishment of the full digital workflow

Patients who visited the Department of Dental Implantology with teeth missing in the anterior maxilla were included in the database of complex esthetic cases if his/her residual ridge displayed a severely collapsed contour and CBCT exam reveals a serious vertical and/or horizontal bone defect. The patients expressed an urgent need for esthetic restoration and normal pronunciation and declared that a removable or fixed teeth-supported prosthetic was unacceptable. The clinician deemed the available bone seriously inadequate for implant placement and aesthetic restoration, making it a must to perform bone and/or soft tissue augmentation first. These complex cases were used to establish the multi-step full digital workflow for esthetic implant therapy. Four major steps were taken:

**Step 1**. Establish the ideal final prosthetic virtually by aligning 3D intraoral scans with facial photographs. The technician virtually fits the restoration and modifies the shape according to specific facial reference and incisal guide on the 3D virtual articulator. The virtual goal will be easily visualized and function as the reference for bone augmentation, implant placement and multiple processes.

**Step 2**. Make a bone augmentation plan virtually. Align the ideal prosthetic, intraoral scans and alveolar bone. Measure the volume of predicted bone augmentation by virtually placing the dental implants and performing bone augmentation. Determine the volume and distribution of bone graft and choose optimal bone augmentation technique.

**Step 3**. Perform digitally guided bone augmentation. Print a surgical guide or a 3D titanium mesh based on the virtual bone augmentation contour. Augment the ridge under the guidance of the surgical guide or using the 3D titanium mesh.

**Step 4**. Make digital measurement at every revisit following bone augmentation and perform -tissue correction timely if needed. The revisit timepoints includes 3 months

(T1, measurement of initial graft resorption), 6 months (T2, implants placement and second bone augmentation if needed), 12 months (T3, soft tissue augmentation if needed and soft tissue conditioning) and 24 months (T4, revisit after final restoration and tissue augmentation if needed) after bone augmentation. Digitally analyze the changes in hard and soft tissue contour by superimposing intraoral scans and bone contour at different follow-up revisit with reference to the initial esthetic prosthetic and bone augmentation goal. Perform additional bone augmentation or soft tissue augmentation to correct the defect if unpredicted graft resorption occurs.

## Teaching methods

All students participated in class sessions on six topics on implant therapy at atrophic anterior maxilla, namely "1. Principle of implant therapy at anterior maxilla", "2. Pre-surgical assessments and treatment plan", "3. Bone augmentation techniques", "4. Implant placement", "5. Soft tissue conditioning" and "6. Management of complications". The curriculum of implant therapy at atrophic anterior maxilla was completed in 12 sessions, with each session lasting for eighty minutes. In the control group, students took courses in turn in the form of teacher-centered lectures, with the teacher's role being to distribute knowledge in the final form. There was no scheduled discussion time during or after the class. In the traditional PBL-CBL group and digital PBL-CBL group, students were divided into teams of 3 or 4 members. The topic was discussed at each of the 24 sessions. The role of the teacher changed from traditional authority to a case narrator and expert guide to discussion. The medical information and the main problems introduced were the same in the two PBL-CBL groups, such as the past medical history, clinical manifestations, X-ray examination results, diagnosis and treatment processes. PBL and CBL were combined as described in our previous study (*Liu et al., 2020*). The difference between the two group is that the problems and cases were expressed in abstract words and clinical pictures in the traditional PBL-CBL group, while those in digital PBL-CBL group were equipped with the established digital workflow, which includes multiple digital virtual animation, measurable data calculation and objective analysis. All courses contain the same knowledge points, and all points are repeated and emphasized to the same degree. The total duration and number of sessions were the same for all three groups.

## Evaluation methodology

Evaluate the efficiency of different teaching methods from the following three aspects. Two teachers in the implant department graded the exam. The name of the students and the group they belong to will be kept secret from the graders.

### Anonymous questionnaire

To evaluate the subjective opinions of the students on different teaching methods, anonymous questionnaires composed of eight questions were filled out by all the students after the training. The detailed information of the questionnaire was shown in Table 1.

### Theory test

At the end of the training, the students took the final exam that consisted of seven questions, namely preoperative assessments of the atrophic ridge, indications for different bone

**Table 1  Rate of satisfaction with the three teaching methods.**

| Items surveyed | Rate of satisfaction | | | P value |
|---|---|---|---|---|
| | Digital PBL-CBL | Traditional PBL-CBL | Control | |
| 1. I like this approach | 94.4% | 83.3% | 77.8% | – |
| 2. This approach is efficient | 88.9% | 66.7% | 44.4% | 0.018 |
| 3. This approach decreases extracurricular work | 55.6% | 55.6% | 83.3% | 0.131 |
| 4. This approach makes learning more targeted and more interesting | 94.4% | 61.1% | 38.9% | 0.002 |
| 5. This approach enhances my ability to analyze and solve problems | 94.4% | 66.7% | 44.4% | 0.005 |
| 6. This approach helps me master theoretical knowledge | 88.9% | 55.6% | 61.1% | 0.070 |
| 7. This approach helps me improve clinical skills | 94.4% | 44.4% | 33.3% | <0.001 |
| 8. This approach facilitates clinician-patient communication | 88.9% | 50.0% | 16.7% | <0.001 |

augmentation techniques, principle of guided bone regeneration, characteristics of different bone graft material, indications of soft tissue augmentation, soft tissue conditioning using interim restoration, and management of complications. The total score was 70 points, 10 points for each question.

*Case analysis*

Data was collected as previously described in (*Liu et al., 2020*). Specifically, after the theory test, the teacher randomly selected a new case with the medical information including the past medical history, clinical manifestations, X-ray examination results and asked the students to answer a series of questions about the diagnostic and therapeutic plan using the key points taught or discussed in class within a limited time in a written form. Finally, the papers were graded anonymously. The total score was 30 points, 6 points for each question.

## Statistical analysis

Pearson's chi-squared test was used to analyze the students' opinions about the teaching methods among the three groups. Scores for the theory test and case analysis were expressed as mean value ± standard deviation (SD). The Shapiro Wilk test was used to determine whether the data are normal distribution. The Kruskal-Wallis H was applied to analyze the difference in scores among the three groups. The Kruskal-Wallis test was used for subsequent pairwise comparison between different groups. All tests were two-sided, with $p < 0.05$ considered significant. SPSS version 26.0 was used to analyze the data.

## RESULTS

### The clinical database and the complex esthetic cases used in digital PBL-CBL method

Twenty-six cases with one to six teeth missing in the anterior maxilla were included as the clinical information database for digital PBL-CBL method. In fifteen cases, 3D printed titanium meshes were fabricated for guided bone regeneration. Four patients received onlay bone block grafting under the assistance of 3D printed surgical guide. Eighteen patients

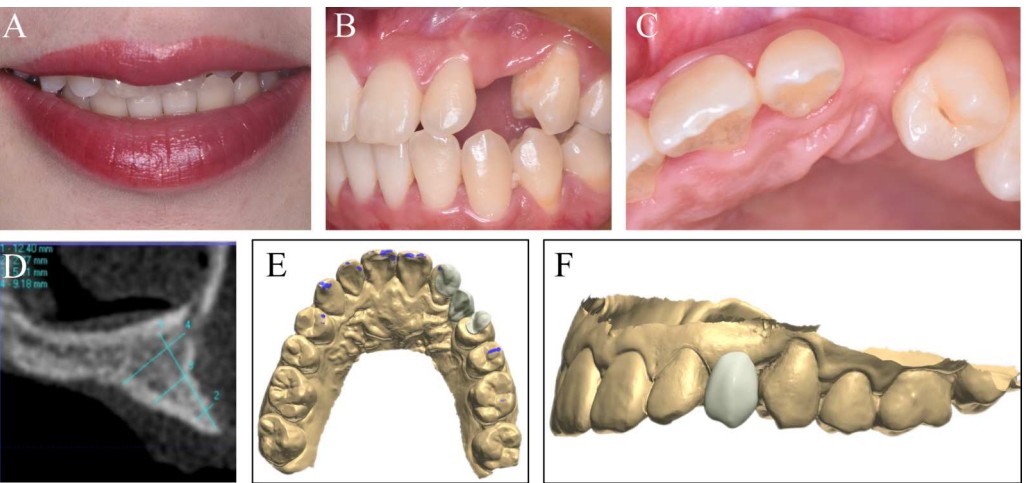

**Figure 1 The preoperative clinical data and CAD fabrication of the ideal prosthetic.** (A) The patient presented a normal smile line. (B) The frontal view and (C) the occlusal view of the residual ridge. (D) Preoperative CBCT revealed severely resorbed alveolar bone. (E) The occlusal view and (F) the frontal view of the CAD design.

underwent second bone augmentation simultaneously with dental implant placement at T2. Six patients received connective tissue grafting before soft tissue conditioning. These cases can cover all the techniques and theoretical knowledge in courses of esthetic implant therapy. A case of guided bone regeneration under the guidance of 3D printed titanium mesh was displayed below to show the digital workflow in detail.

The patient was a 26-year-old female patient who had #23 tooth extracted 5 years ago. A clinical exam revealed good oral hygiene and a normal smile line (Figs. 1A, 1B and 1C). The frontal and occlusal view of the residual ridge displayed a severely collapsed contour and gingival recession at #24 tooth (Figs. 1B and 1C). The preoperative CBCT reveals a severely vertical and horizontal bone defect with a bladed alveolar crest (Fig. 1D).

**Step 1**. The first step was to create an ideal final prosthetic. The shape and form of the teeth were modified according to the facially generated esthetic principles and occlusal function (Figs. 1E and 1F).

**Step 2**. The preoperative CBCT DICOM data of the jaw was imported into Mimics Research 21.0 (Materialise, Leuven, Belgium) for 3D reconstruction (Fig. 2A). The aesthetic prosthetic established in Step 1, the dental implant model, and the reconstructed 3D model of jaw bone were transferred into 3-matic Research 13.0 (Materialise, Leuven, Belgium) for simulation of restoration-oriented implant placement (Fig. 2B). The alveolar bone defect was virtually reconstructed considering the minimal bone tissue needed surrounding the implant, the ideal alveolar bone contour and soft tissue condition. The augmented bone surface was then over-thickened to reconstruct the horizontal and vertical contour, establishing the bone augmentation goal that matches the specific prosthetic goal (Fig. 2C). Based on the virtual reconstruction of hard tissue, the titanium mesh for bone augmentation was designed and produced (Figs. 2D and 2E).

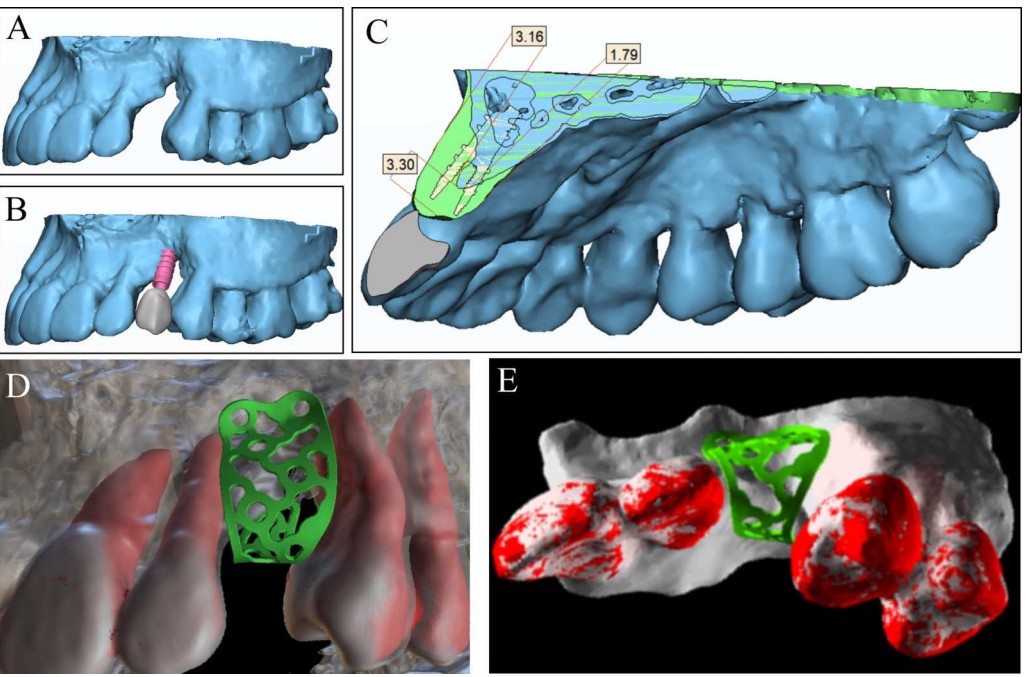

**Figure 2  Establishing bone augmentation plan by designing a titanium mesh.** (A) The reconstructed 3D model of the preoperative maxilla. (B) Align the aesthetic prosthetic model with the preoperative maxilla model and virtually place the dental implants in a restoration-oriented way. (C) Virtually reconstruct the alveolar bone defect according to the bone volume required by the dental implant. (D–E) Design a titanium mesh model accurately mirroring the contour of ideal bone dimension.

**Step 3**. The titanium mesh was tried on and adjusted to the optimal position (Figs. 3A and 3B). Particulate autogenous bone chips and deproteinized bovine bone were mixed up with injectable platelet-rich-fibrin. The space between the titanium mesh and bone surfaces was compactly filled with the bone graft (Fig. 3C). The wound was closed with no tension (Fig. 3D).

**Step 4**. Result-focused analysis and timely-correction were performed at 3 months (T1), 6 months (T2), 12 months (T3), and 24 months (T4) after bone augmentation. At T1, the augmented ridge contour digitalized by intraoral scans was superimposed on the preoperative contour, showing vertical tissue collapse mesial to #24 tooth (Figs. 4A and 4B) although superimposing of bone contour on the original condition revealed significant improvement in bone volume (Fig. 4C). When superimposed on the ideal bone contour, the bone contour at T1 failed to match the goal on the top of the ridge, leaving a vertical bone defect of 2.47 mm in height (Fig. 4D).

Six months after the first bone augmentation (T2), soft tissue resorption continued that the titanium mesh can be seen through the overlying gingiva (Fig. 4A). Superimposition of ridge contour revealed more vertical and horizontal contour collapse compared with that at T1 (Fig. 4B). Superimposition of bone contour on the original (Fig. 4C) and ideal bone condition (Fig. 4D) showed improvement in horizontal bone volume after grafting but unmet vertical bone dimension requirement at T2, which required an additional

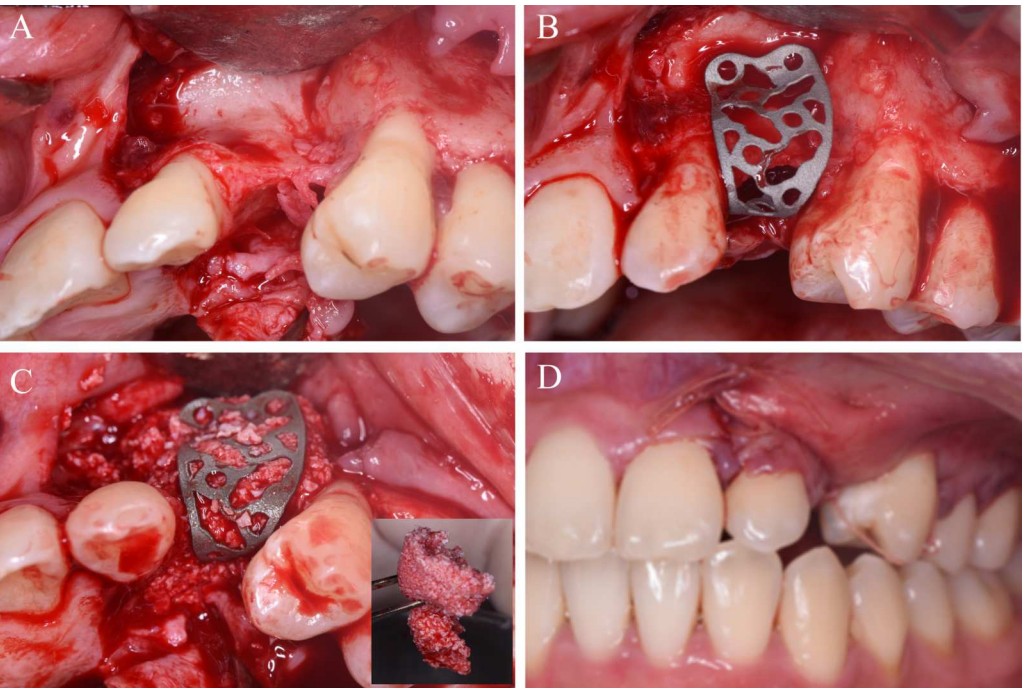

**Figure 3    The first bone augmentation using 3D printed titanium mesh.** (A) Flap elevation revealed the severely resorbed bone contour. (B) Set the 3D printed titanium mesh in place. (C) The space between the titanium mesh and bone surface was compactly filled with the sticky bone graft, a mixture of particulate autogenous bone, deproteinized bovine bone, and injectable platelet-rich-fibrin. (D) Close the wound with no tension.

bone augmentation. The clinician deemed guided bone regeneration simultaneously with implant placement be suitable for the patient (Fig. 5). Superimposition of the actual and the planed location of implants showed good accuracy of the guided surgery (Figs. 5G, 5H and 5I).

The ridge contour seemed to be vertically improved 6 months after the corrective surgery (T3) (Figs. 4A and 4B), although a minor horizontal bone defect remained distal to #22 tooth when superimposed on the ideal contour (Fig. 4D). The clinician decided there was no need for additional surgical correction as the abundant vertical bone volume suggested complete filling of the interproximal space after restoration and the minor horizontal contour defect could be corrected by soft tissue conditioning using provisional crown (Fig. 5J). Three months after soft tissue conditioning, the provisional was replaced with the final restoration, and the contour of the ridge was greatly improved as viewed intraorally (Fig. 5K). Good esthetic result was achieved as viewed intraorally (Fig. 5L). Nine months after final restoration (T4), X-rays revealed no obvious resorption in peri-implant bone level. Superimposition of ridge contour showed stable and abundant bone volume (Figs. 4A and 4B), suggesting the aesthetic goal established in Step 1 was realized.

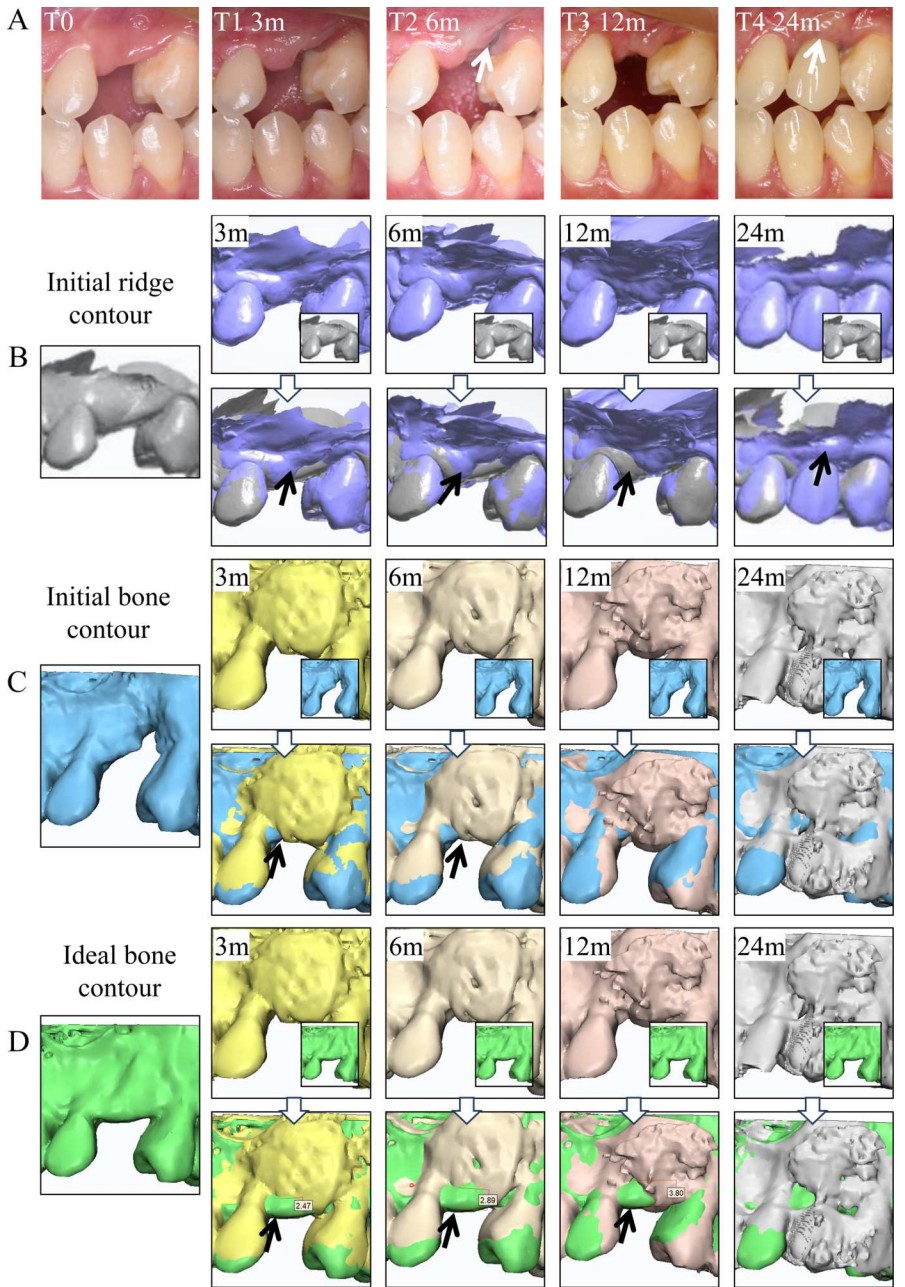

**Figure 4 The digital superimposition of ridge contour at different time points to accurately evaluate the condition of tissue resorption.** (A) The intraoral frontal views of the residual ridge preoperatively (T0), and at 3 months (T1), 6 months (T2), 12 months (T3), and 24 months (T4) after the first bone augmentation. (B) Superimposing the intraoral scans at T1, T2, T3, and T4 on the initial ridge contour. (C) Superimposing the bone contour at T1, T2, T3, and T4 on initial bone contour. (D) Superimposing the bone contour at T1, T2, T3, and T4 on the ideal bone contour.

## Digital PBL-CBL vs traditional PBL-CBL/didactic teacher-centered method

A total of 54 students (30 men and 24 women), aged between 23 and 27 years (mean: 24.6 years) were included in this study. All the students kept up with the class sessions. There

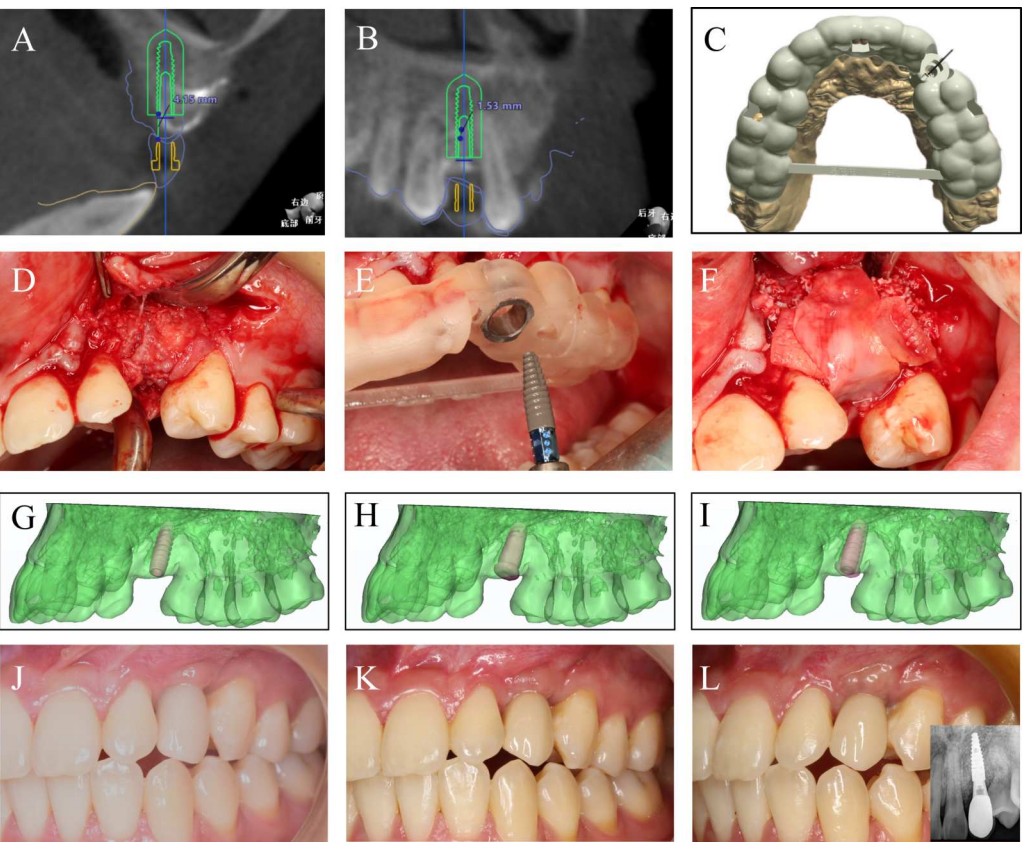

**Figure 5** **The second bone augmentation and dental implant placement using implant surgical guide.** (A–C) An implant surgical guide was designed based on the prosthetic goal established in Step 1. (D) The augmented ridge, showing good horizontal bone contour and compact bone quality. (E) Placing implant as designed. (F) Second guided bone augmentation. (G) The planed location of the implant. (H) The actual location of the implant. (I) Superimposition of G and H revealed high reliability of the surgical guide. (J) Provisional restoration for soft tissue conditioning. (K) Final restoration. (L) Revisit 24 months after the first augmentation surgery, showing good aesthetics and stable peri-implant bone level.

was no significant difference among the three groups with regard to gender and age. All students adhere to the schedule and attend lectures or discussions on time.

Table 1 shows the rate of satisfaction with different teaching methods. Students in the traditional PBL-CBL group reported higher rate of satisfaction than those in control group in all items except for "This approach decreases extracurricular work" and "This approach helps me master theoretical knowledge". The traditional PBL-CBL method seemed to increase extracurricular work and showed no advantage in delivering theoretical knowledge according to the students' opinions. With the assistance of problems and cases expressed in a full-digital way, the digital PBL-CBL method resulted in a higher rate of satisfaction than the traditional PBL-CBL method and the didactic teacher-centered method in all items except for "This approach decreases extracurricular work". Students reported similar opinions on the amount of the extracurricular work in the traditional PBL-CBL and digital PBL-CBL groups. Statistical significance between the two groups was

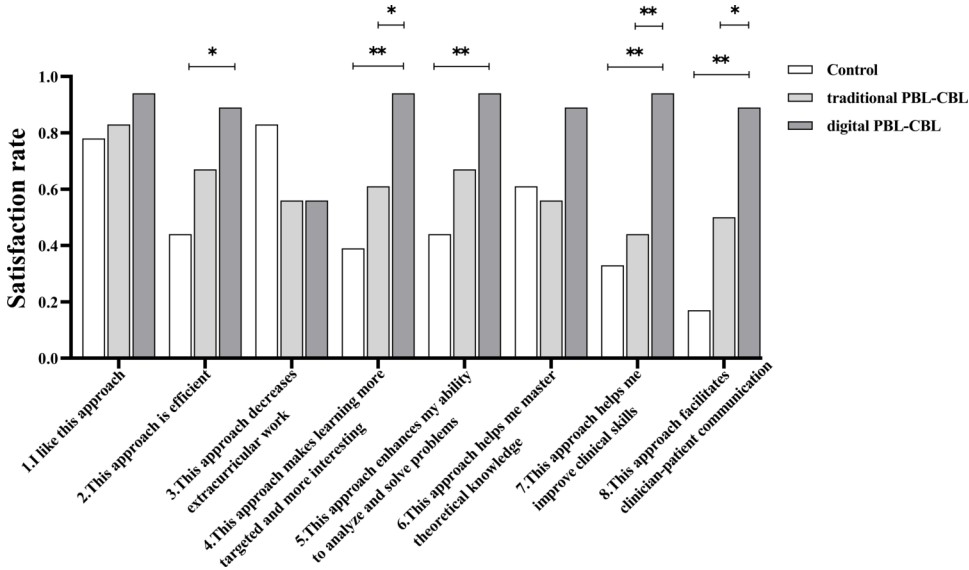

**Figure 6** Rate of satisfaction with the three teaching methods. $**P < 0.01$, $*P < 0.05$.

detected for items "This approach makes learning more targeted and more interesting", "This approach helps me improve clinical skills" and "This approach facilitates clinician-patient communication" (Fig. 6), suggesting digital assistance amplifies the advantages of traditional PBL-CBL method and benefits practical thinking and clinical problem-solving abilities according to the subjective opinions of students. Statistical significance between the control group and the digital PBL-CBL group was detected for items "This approach is efficient", "This approach makes learning more targeted and more interesting", "This approach enhances my ability to analyze and solve problems", "This approach helps me improve clinical skills" and "This approach facilitates clinician-patient communication" (Fig. 6), suggesting compared with the didactic teacher-centered method, the digital PBL-CBL method improves students' subjective initiative, self-learning skills, clinical skills and communication skills according to the subjective opinions of students.

Table 2 shows the scores of the theory test and the case analysis. Traditional PBL-CBL method resulted in higher scores in theory test and case analysis than the didactic teacher-centered method, although the differences achieved no statistical significance (Fig. 7). With digital assistance, the advantages of the traditional PBL-CBL method in upgrading scores were further improved, with the difference between the digital PBL-CBL method and the didactic teacher-centered method being significant in both theory test and case analysis (Fig. 7). Particularly, digital PBL-CBL method led to significantly higher scores than the traditional PBL-CBL method in case analysis (Fig. 7), suggesting digital assistance benefits master of practical thinking and clinical problem-solving abilities. When theory test and case analysis were evaluated as total score, digital PBL-CBL method showed obvious advantages over the didactic teacher-centered method and the traditional PBL-CBL method, while no significant difference was detected between the control group and the traditional PBL-CBL group.

**Table 2  Comparison of average scores of the three groups ($n = 18$).**

| Group | Theory test | Case analysis | Total score |
|---|---|---|---|
| Digital PBL-CBL group ($n = 18$) | $61.00 \pm 6.80$ | $20.06 \pm 2.98$ | $81.06 \pm 8.05$ |
| Traditional PBL-CBL group ($n = 18$) | $55.22 \pm 9.86$ | $14.06 \pm 5.47$ | $69.28 \pm 9.39$ |
| Control group ($n = 18$) | $45.11 \pm 12.76$ | $13.83 \pm 4.60$ | $58.94 \pm 14.81$ |
| Kruskal-Wallis H | 14.220 | 16.863 | 21.387 |
| $P$ | <0.01 | <0.01 | <0.01 |

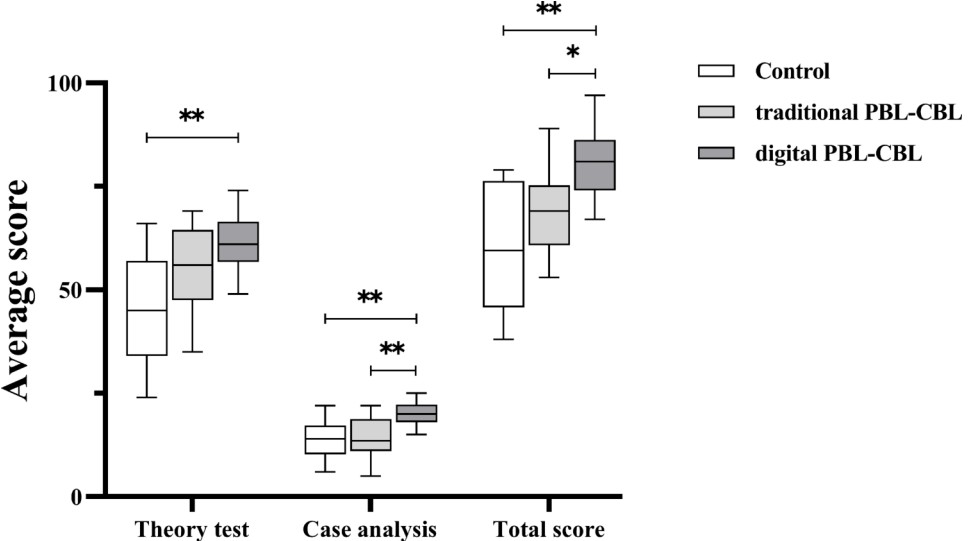

**Figure 7  Score of theory test, case analysis and the two combined.** $^{**}P < 0.01$, $^{*}P < 0.05$.

## The extra concrete examples of digital PBL-CBL teaching method

There have been two graduate students who have been trained by digital PBL-CBL method won the first prize and the third prize, respectively in the 12th and 10th BITC Finals for Implant Dentistry (BITC, Beijing Implant Training College, a platform dedicating to training/education and research in dental implantology). This competition is divided into four themes, namely bone augmentation theme, aesthetic area implant therapy theme, dentition loss implant therapy theme and digital implant therapy theme. One of the students won the first prize in "The Bone Augmentation Theme Competition of the 12th BITC Finals for Implant Dentistry", another student won the third prize in "The Digital Implant Therapy Theme Competition of the 10th BITC Finals for Implant Dentistry". The competition requires participants to present clinical studies or case reports of implant dentistry through PowerPoint presentation within a specified time. The ability to critically think, analyze and solve real problems and their communication skills have been cited as the scoring criteria of the competition. The two graduate students trained with digital PBL-CBL method but none in non-digital PBL-CBL group achieved excellent results in the BITC National Finals, illustrating the effectiveness of the digital PBL-CBL method.

## DISCUSSION

Although the application of digital tools is widespread in routine dental care, this trend towards digitization has not been extended to dental curricula, making it difficult to prepare future dentists for digital work-life (*Zitzmann et al., 2020*). The traditional PBL-CBL method has been described as a promising tool for leadership training, biochemistry experiment teaching, international classification of diseases encoding and dental education (*Dong & Zeng, 2017*; *Ginzburg et al., 2018*; *Liu et al., 2020*; *Yang et al., 2023*). To maximize the efficiency of traditional PBL-CBL method, we combined PBL and CBL using fully digital clinical cases in teaching implant therapy at atrophic anterior maxilla. According to the students' feedback, digital PBL-CBL method won over traditional PBL-CBL method as significantly more students believed that digitalization made learning more efficient and targeted, enhanced their problem-solving ability, improved their clinical skills and facilitated communication with patients. This superiority was echoed by higher scores of theory test and case analysis in digital PBL-CBL group, proving that digital PBL-CBL method was more effective than the traditional PBL-CBL method in teaching complex dental implant therapy.

Previous research has pointed out that didactic teacher-centered approach was more effective than CBL in conveying existing knowledge system (*Allchin, 2013*; *Jamkar, Yemul & Singh, 2006*). As PBL is able to cover about 80% of the work that could be accomplished in a contemporaneous didactic approach (*Albanese & Mitchell, 1993*; *Berkson, 1993*), we combined CBL with PBL by interrupting cases with a series of well-contextualized questions or problems in teaching maxillary sinus floor elevation surgery, and proved that the combined PBL-CBL method (the traditional PBL-CBL method) was even advantageous over the didactic approach in conveying existing knowledge system (*Liu et al., 2020*). In this study, however, the traditional PBL-CBL method showed no advantage over the didactic method in delivering theoretical knowledge when used to teach complex surgeries at atrophic maxilla according to the students' opinions and theory test. This conflict may be attributed to the different curricular contents as the surgical and prosthetic manipulation at atrophic maxilla is more complex and flexible compared with maxilla sinus lifting, which requires more solid knowledge, vivid imagination and greater observation abilities. When cases or problems were contextualized with digital hardware and software, the efficiency of traditional PBL-CBL method in getting across standard curricular content was significantly enhanced as more students (88.9%) believed that digital PBL-CBL method helps them master theoretical knowledge than the traditional PBL-CBL method (55.6%) or the didactic approach (61.1%), and the digital PBL-CBL method ($61.00 \pm 6.80$) scores higher than the traditional PBL-CBL method ($55.22 \pm 9.86$) or the didactic teacher-centered method ($45.11 \pm 12.76$) in theory test. Two graduate students won prizes in BITC National Finals, revealing the digital PBL-CBL method could help dental students acquire effective information quickly in a limited amount of time, as well as enable them to think actively and guide them to summarize essential information during the learning process. The establishment of the clinical database, which is a feature that clearly distinguishes digital PBL-CBL method from didactic teacher-centered method,

has established real medical scenes and encouraged students to adopt a proactive attitude towards learning toward shifting from passive acceptance paradigm to active acquisition. These results suggested that over-all digitalization of clinical problems and cases could help cover standard curricular content and assist students to form a comprehensive and organized understanding of esthetic implant therapy.

Case analysis was designed to test students' ability in practical thinking, careful observation, and flexible decision making. Traditional PBL-CBL method was proved to raise scores of case analysis about maxillary sinus lifting when compared with the didactic teacher-centered method (*Liu et al., 2020*). However, in this study about esthetic implant therapy, it showed no superiority to the didactic teacher-centered method in case analysis, indicating problems or cases explained in traditional words or pictures may not be potential enough to manage teaching task of high complexity and flexibility. The major difficulty in restoring missing teeth at atrophic anterior maxilla is to augment the ridge according to the 3D position of final prosthetics and make timely correction in case of unpredicted tissue collapse following augmentation. In traditional PBL-CBL method, the students are usually told directly by the teacher when and how to perform a certain surgical or prosthetic technique through presentation of words or pictures, rather than making the critical decision through careful observation and calculation by themselves. With the assistance of digital hardware and software, students can preview the 3D bone contour, the soft tissue contour and the ideal final prosthetic at any moment during the treatment period. By superimposing myriad scans and precisely aligning them with common data points, students can calculate the amount of tissue augmentation accurately guided by the 3D position of the ideal final prosthetic and make timely correction when contour discrepancies appeared due to tissue resorption. In our study, the results of case analysis showed that students in the digital PBL-CBL group were more likely to use the acquired knowledge spontaneously to solve new problems than those who acquired the same information through lectures or traditional PBL-CBL method, proving the digital tools in teaching would stimulate students' imagination, practice their observation ability, and sharp their logic based on objective data.

Problems or cases explained with a full digital workflow may provide the students a chance to view clinical cases in a more accurate, measurable and intuitive way. However, there is seldom a purely conventional pathway or a fully digital workflow in routine dental practice (*Joda & Brägger, 2016*; *Kapos & Evans, 2014*). There are some limiting factors in the application of the digital workflow. Firstly, quite a few factors would make it difficult to superimpose diverse tissue structures and guarantee the overall precision of superimposition, including the choice and visibility of the defined reference points, the power of the software algorithm, and the corresponding digital data obtained from radiology and scanning techniques (*Plooij et al., 2011*). Besides, the purchase, installation, facilities set-up, updating, and maintenance of the equipment are expensive meanwhile the implementation of new technology is time consuming and requires the operator's patience for an individual learning curve (*Holden & Karsh, 2010*; *Joda & Brägger, 2015*; *van der Zande, Gorter & Wismeijer, 2013*). When implementing the digital workflow in daily life, new treatment options must be trained and a learning curve must be considered. The

correct indication of the digital method and its application are the premise and crucial to the success of the whole process. This requires a teamwork approach and could affect clinicians, dental assistants and technicians.

There were several limitations in our study. First, in order to get feedback on the new hybrid approach, Likert scales are more suitable and accurate to measure responses and find differences than Yes/No scales that can only perform rough calculations. Second, we analyzed results from only one clinical department within our institution. These results may have been different beyond our institution. Third, the sample size of students was relatively small, which caused difficulties in statistical analysis, especially for the anonymous questionnaire. Further randomized controlled trial with large sample size was needed to confirm the efficiency of the digital PBL-CBL method.

## CONCLUSION

In conclusion, compared with the traditional PBL-CBL method and the didactic teacher-centered method, the digital PBL-CBL method may be effective for improving the level of the students' theoretical knowledge and enhancing their case analysis skills and capabilities when learning complex implant cases at atrophic anterior maxilla. The digital PBL-CBL method provided a promising alternative for teaching complex implant cases at the anterior maxilla. In the implementation process of the digital PBL-CBL method, teachers should adapt the teaching method and students should adjust their attitudes towards learning. The successful application of the digital PBL-CBL method will not improve overnight, which requires the full participation of teachers and students and the provision of timely feedback so that positive adjustments can be made continuously.

### Funding
This work was supported by the Scientific Research Project of Chongqing Municipal Commission of Health (2022QNXM005) and the Postgraduate Education Reform Research Project of Chongqing Medical University (xyjg220209). The funders had no role in study design, data collection and analysis, decision to publish, or preparation of the manuscript.

### Grant Disclosures
The following grant information was disclosed by the authors:
Scientific Research Project of Chongqing Municipal Commission of Health: 2022QNXM005.
Postgraduate Education Reform Research Project of Chongqing Medical University: xyjg220209.

### Competing Interests
The authors declare there are no competing interests.

## Author Contributions

- Dan Chen performed the experiments, authored or reviewed drafts of the article, and approved the final draft.
- Wenyan Zhao performed the experiments, analyzed the data, prepared figures and/or tables, authored or reviewed drafts of the article, and approved the final draft.
- Li Ren performed the experiments, analyzed the data, prepared figures and/or tables, and approved the final draft.
- Kunli Tao performed the experiments, prepared figures and/or tables, and approved the final draft.
- Miaomiao Li performed the experiments, analyzed the data, prepared figures and/or tables, and approved the final draft.
- Beiju Su analyzed the data, prepared figures and/or tables, and approved the final draft.
- Yunfei Liu analyzed the data, authored or reviewed drafts of the article, responsible for the establishment of clinical case database and human sample collection, and approved the final draft.
- Chengzhe Ban conceived and designed the experiments, prepared figures and/or tables, authored or reviewed drafts of the article, and approved the final draft.
- Qingqing Wu conceived and designed the experiments, authored or reviewed drafts of the article, and approved the final draft.

## Human Ethics

The following information was supplied relating to ethical approvals (i.e., approving body and any reference numbers):

The Ethics Committee of the Stomatological Hospital of Chongqing Medical University granted ethical approval to carry out the study within its facilities (Ethical Application Ref: KQJ2022166).

## Data Availability

The raw measurements are available in the Supplementary Files.

## Supplemental Information

Supplemental information for this article can be found online at http://dx.doi.org/10.7717/peerj.16496#supplemental-information.

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
