# Peer review of "Digital PBL-CBL teaching method improves students’ performance in learning complex implant cases in atrophic anterior maxilla"

_PeerJ, doi:10.7717/peerj.16496_

## Round 0.1 · original submission · Major Revisions

Dear author,

Kindly take into account the revisions proposed by the reviewers.

Sincerely,

Reviewer 1 ·

Basic reporting

No comment

Experimental design

no comment

Validity of the findings

the article can improve the empirical evidence to support its claims. While the authors mention positive outcomes and improved student performance, they could improve by adding extra concrete examples, case studies, or even qualitative feedback from students would have strengthened the credibility of the article and provided a more convincing argument for the effectiveness of the digital PBL-CBL method.

Additional comments

The writing style of the article is generally clear and concise, making it accessible to readers within the field of dental implant surgery. However, there are instances where technical terms and jargon are used without sufficient explanation, which may hinder the understanding of non-expert readers. The authors could have provided more context or definitions for specialized terms to ensure clarity for a wider audience.

·

Basic reporting

English was not so clear. Please see the attached revised version of the main text. The Literature, figures, and tables are ok. There were 27 points of plagiarism (suggested to rephrase)

Experimental design

In spite of PBL and CBL being acknowledged, I suggested explaining what PBL/CBL stands for.

Validity of the findings

Clear, but not thar robust.

Reviewer 3 ·

Basic reporting

Dear Authors,

I have meticulously reviewed your manuscript entitled "Application of digital PBL-CBL method in teaching dental implant surgery at atrophic anterior maxilla". I recognize the importance of your research, yet believe that several aspects could be enhanced for greater clarity and coherence. Here are my recommendations:

1. *Title Revision:* Consider revisiting the title to more accurately reflect the core objective of your study.
2. *Introduction Streamlining:* I would recommend condensing the section "To prepare the dental students for such high clinical demands, it is necessary for the dental schools to improve quality of classroom teaching of dental implant therapy at severely atrophic maxilla", as it seems a bit too prescriptive. Try to focus on presenting the existing gap in the literature which your study aims to address.
3. *Reframing the Introduction:* Please abstain from including personal opinions in the introduction. A more objective and grounded approach would strengthen this section.
4. *Materials and Methods Section:* Begin this section with details on the students and groupings to provide readers with a clearer understanding of the study context.
5. *Discussion Section Enhancement:* Incorporate a paragraph within the discussion section detailing the limiting factors of the digital technique workflow to give a more rounded view of the research.
6. *Conclusion Section:* Kindly expand the conclusion to encompass some of the potential limitations of the technique discussed in the study.
7. *Reference Update:* Ensure that the references are up-to-date to substantiate your manuscript with the most recent and relevant literature.

Best regards,
Reviewer

Experimental design

No comment

Validity of the findings

No comment

Additional comments

No comment

---

## Round 0.2 · accepted · Accept

Thank you for submitting the revised version of your manuscript, which has resulted in significant improvements to the quality of the study. I reviewed the manuscript and I am satisfied with the modifications provided by the authors.